# Diabetic Kidney Disease, Cardiovascular Disease and Non-Alcoholic Fatty Liver Disease: A New Triumvirate?

**DOI:** 10.3390/jcm10092040

**Published:** 2021-05-10

**Authors:** Carolina M. Perdomo, Nuria Garcia-Fernandez, Javier Escalada

**Affiliations:** 1Department of Endocrinology and Nutrition, Clínica Universidad de Navarra, 31008 Pamplona, Spain; cperdomo@unav.es (C.M.P.); fescalada@unav.es (J.E.); 2IdiSNA (Instituto de Investigación en la Salud de Navarra), 31008 Pamplona, Spain; 3Department of Nephrology, Clínica Universidad de Navarra, 31008 Pamplona, Spain; 4Red de Investigación Renal (REDINREN), Instituto de Salud Carlos III, 28029 Madrid, Spain; 5CIBERObn (CIBER Fisiopatología de la Obesidad y Nutrición), Instituto de Salud Carlos III, 28029 Madrid, Spain

**Keywords:** diabetic kidney disease, cardiovascular disease, non-alcoholic fatty liver disease

## Abstract

Non-alcoholic fatty liver disease is a highly prevalent disease worldwide with a renowned relation to cardiovascular disease and chronic kidney disease. These diseases share a common pathophysiology including insulin resistance, oxidative stress, chronic inflammation, dysbiosis and genetic susceptibilities. Non-alcoholic fatty liver disease is especially prevalent and more severe in type 2 diabetes. Patients with non-alcoholic fatty liver disease should have liver fibrosis assessment in order to identify those at the highest risk of adverse outcomes so that appropriate management strategies can be implemented. Early diagnosis and treatment of non-alcoholic fatty liver disease could ameliorate the burden of cardiovascular disease and chronic kidney disease.

## 1. Introduction

Nonalcoholic fatty liver disease (NAFLD) is a liver condition defined as the accumulation of intracellular fat in hepatocytes in the absence of excessive alcohol intake, certain metabolic diseases or pharmacological treatment [1,2]. Although NAFLD is considered the liver component of the metabolic syndrome (MS), progression to liver fibrosis has been associated with type 2 diabetes mellitus (T2D) [3], cardiovascular disease (CVD) [4], chronic kidney disease (CKD) [5] and other comorbidities. NAFLD is increasingly common in the western world, and it is especially prevalent (50–70%) and severe in T2D patients [6]. Early diagnosis and treatment of NAFLD could ameliorate the burden of CKD and CVD. Due to the invasiveness of liver biopsy, liver elastography (LE) and non-invasive liver fibrosis markers (e.g., NAFLD Fibrosis Score (NFS) [7], Fibrosis 4 Score (FIB-4 Score) [8] and Hepamet Fibrosis Score (HFS) [9]) have been proposed as high diagnostic precision tools for advanced histological stages of fibrosis [10]. Thus, it is important not only to recognize NAFLD and differentiate progressive and non-progressive NAFLD patients, but also to be aware of the severity and multidisciplinary nature of this disease. The aim of this review is to raise awareness about the relationship of NAFLD, CVD and CKD in T2D patients and to improve our understanding and assessment of those conditions. This review intends to give a general approach to these three diseases and not a specific approach for each condition.

## 2. Non-Alcoholic Fatty Liver Disease and Cardiovascular Disease

The presence and severity of NAFLD is strongly associated with a higher mortality from any cause [4] but mainly cardiovascular death due to an increased risk of subclinical atherosclerosis and major cardiovascular events (MACE) [4,11,12]. In 2016, a metanalysis (*n =* 34,043 patients; 36.3% with NAFLD) showed more MACE in advanced stages of NAFLD [4]. Additionally, in a large US database (*n* = 55,099,280 patients), NAFLD was associated with acute myocardial infarction (odds ratio, OR 1.5; 95%, confidence interval CI: 1.40–1.62) even after adjusting for traditional risk factors [12]. Similarly, in a large sample of a general population (*n* = 8511), after a mean follow-up of ten years, liver fibrosis (assessed through the FIB-4 Score) was independently associated with CVD even after adjusting for other risk factors such as sociodemographic characteristics, the Systematic Coronary Risk Evaluation calculator (SCORE), use of statins and use of aspirin (hazard ratio, HR 1.63; 95% CI: 1.29–2.06) [12]. Ballestri et al. compared the diagnostic performance of various liver fibrosis biomarkers in chronic liver disease (viral hepatitis and NAFLD), demonstrating for FIB-4 Score, Forns index, NFS and HFS a positive correlation with cardiovascular risk scores (SCORE or the Framingham risk scoring systems) [13].

Other associations between NAFLD and CVD have been described, such as the degree of steatosis per se with the incidence of CVD [14], the increased risk of atrial fibrillation [15] and the significantly higher risk of all-cause mortality in hospitalized patients with CVD and NAFLD vs. non-NAFLD patients (adjusted HR 2.08; CI 95% 1.56–2.59; *p* < 0.001) [16]. Therefore, it is clear that NAFLD seems to improve the identification of high-risk CVD patients.

Regarding T2D, higher liver fat content and liver fibrosis may predict worse cardiovascular risk. In fact, Ichikawa et al. found that NAFLD, which was assessed through computerized tomography (CT), in addition to coronary artery calcium (CAC) and the Framingham risk score, was useful to identify T2D patients at higher risk of CVD (*n* = 529, median follow-up: 4.4 years) [17]. Chun et al. found that the severity of liver fibrosis independently predicted CVD in patients with newly diagnosed T2D (*n* = 1481, median follow-up: 88.1 months) [18]. Regarding subclinical CVD and T2D, a cross-sectional study (*n* = 1878) showed that non-invasive liver fibrosis scores were independently associated with subclinical myocardial remodeling [19]. Epicardial adipose tissue (EAT) may represent an earlier and improved marker of subclinical cardiovascular risk in asymptomatic patients. In the CAESAR (CArdiometabolic risk, Epicardial fat, and Subclinical Atherosclerosis Registry) (*n* = 2277), both EAT (assessed through echocardiography) and the presence of NAFLD were associated with CAC, with a stronger association to CAC with the presence of increased EAT in NAFLD patients [20]. Interestingly, in a recent prospective, cross-sectional study (*n* = 100 diabetic patients) [21], EAT (measured by magnetic resonance (MR)) was higher in NAFLD patients. A recent meta-analysis [22] including 13 case-control studies (*n* = 2260) confirmed that EAT was significantly increased in NAFLD patients. It was also correlated with the severity of hepatic steatosis and fibrosis, and atherosclerotic cardiovascular disease.

In summary, liver fat content and fibrosis may increase cardiovascular risk, so identifying even mild forms of NAFLD in diabetic individuals may justify treatment initiation and risk factor modification to reduce cardiovascular risk.

## 3. Non-Alcoholic Fatty Liver Disease and Chronic Kidney Disease

In the last decade, three meta-analyses have demonstrated not only an association between NAFLD and increased risk of CKD, regardless of the diabetes status, but also a negative impact of NAFLD severity on the prevalence of CKD [23,24] and risk of albuminuria [25]. Likewise, an association between NAFLD and development of advanced CKD (*p* < 0.05) has been described in cohort studies [5,26]. This association did not change after stratifying by age, gender and the presence of other comorbidities such as T2D, obesity, hypertension and ischemic heart disease. Higher CKD prevalence has been described in NAFLD patients with liver fibrosis (22.14% vs. 4.82%, *p* = 0.001) [27]. Regarding the association between non-invasive liver fibrosis scores with CKD, a FIB-4 score higher or equal than 1.100 has been described as the best indicator of CKD (OR 2.660, 95% CI 1.201–5.889; *p* = 0.016) [28].

Although an increase in overall mortality risk has been observed in patients with NAFLD and CKD, it may be explained by the high prevalence of metabolic comorbidities such as T2D [29]. Specifically, a high prevalence of microvascular and macrovascular complications have been evidenced in patients with T2D and liver fibrosis [30]. Diabetic kidney disease (DKD) is the name given to the kidney damage related to diabetes. DKD affects about 40% of patients with T2D [31]. In a cohort of 2103 T2D patients, an increased risk of DKD has been described for patients with NAFLD compared to patients without NAFLD (OR 1.87; 95% CI 1.3–4.1, *p* = 0.020), independent of age, gender, body mass index (BMI), waist circumference, hypertension, diabetes duration, HbA1c, lipids, smoking and medication use [32]. Similarly, in a cohort of 1763 T2D Chinese patients, a higher prevalence of albuminuria was observed in patients with liver steatosis and advanced fibrosis compared to non-NAFLD patients (46.2% vs. 64.2% vs. 41.4%, respectively *p* = <0.001) [33]. Moreover, an increased risk of albuminuria was associated with advanced fibrosis independently of HbA1c, hypertension and BMI (OR 1.52; 95% CI 1.02–2.28; *p* = 0.039).

Regarding the impact of NAFLD on CKD in T2D patients, there is data associating liver steatosis with albuminuria excretion rate (OR 3.49; 95% CI 2.05–5.94, *p* < 0.01) and liver fibrosis with CKD (OR 6.39; 95% CI 4.05–10.08, *p* < 0.01) [34]. Similarly, an observational study including 1108 T2D patients with NAFLD demonstrated an increased prevalence of albuminuria [35]. A higher significant prevalence of liver fibrosis has been described in patients with isolated non-albumin proteinuria and albuminuria than in the non-proteinuria group (18.7% vs. 16.5% vs. 9.5%, *p* = 0.001). As a matter of fact, some authors consider non-albumin proteinuria as a marker of T2D complications. On the other hand, lower levels of glomerular filtration rate (GFR) were associated with increased probability of liver fibrosis (higher NFS) in 2689 patients with NAFLD and T2D [36]. In T2D patients, liver fibrosis could be predicted using GFR levels (AUROC: 0.71, cut-off point: 92.78 mL/min/1.73 m^2^; *p* = <0.001).

There is enough evidence that NAFLD is associated with DKD burden, and NAFLD treatment should have a good impact on kidney function. In fact, an improvement in liver fibrosis stage was independently associated with an increase in GFR values in a post hoc analysis of 261 NAFLD patients who were treated with lifestyle modifications during 52 weeks [37].

Integrating the triumvirate, in 2019, Mantovani et al. [38] analyzed 137 patients with T2D. The presence of CKD (estimated GFR < 60 mL/min/1.73 m^2^ and/or abnormal albuminuria) and increased cardiovascular complications (previous ischaemic heart disease, ischaemic stroke, permanent AF) significantly increased across liver stiffness tertiles (from around 15% in tertile 1 to 45% in tertile 3).

## 4. Common Pathophysiology

Several studies showed the importance of NAFLD and its relationship to CVD and CKD in T2D patients. They all share common risk factors, making it difficult to unravel causal relationships of this triumvirate [39]. Nevertheless, NAFLD seems to be an additional independent risk factor for CVD [1,40,41] and CKD [5,24,26,39,42], and CKD is a major risk factor for CVD [39].

It is known that lipolysis of adipose tissue, dietary lipids and de novo lipogenesis contribute to the pool of lipids stored in the liver [43] and, in a normal situation, insulin is the main mediator of these processes [44]. However, in the presence of insulin resistance (IR), there is an increased plasma concentration of non-esterified fatty acids from ectopic fat. On the other hand, de novo lipogenesis is the process in which hepatocytes convert excess carbohydrates (especially fructose) into fatty acids through an insulin-independent transporter [43]. Liver lipid removal is mediated by mitochondrial fatty acid β-oxidation and re-esterification to form triglycerides, which can be exported into the blood as VLDL or may be stored in lipid droplets [43,45]. Liver triglyceride accumulation enhances IR resulting in compensatory hyperinsulinemia which decreases glycogen synthesis and increases the hepatic fatty acid uptake, alters triglycerides transportation and inhibits liver β-oxidation [45]. All these alterations lead to an activation of hepatic macrophages with a further intensification of the pro-inflammatory cytokine activity (such as tumor necrosis factor-α, interleukin-6 and interleukin-8) leading to oxidative stress-mediated lipotoxicity, increased activity of the renin-angiotensin-aldosterone system (RAS) [35], platelet activation [39], inflammasome activation and mitochondrial dysfunction, contributing to a broaden exacerbation of IR [45], hepatocellular injury and the progressive accumulation of excess extracellular matrix through hepatic stellate cells activation [39,44]. RAS activation has been implied in the production of pro-inflammatory cytokines, aggravating oxidative stress and supporting subclinical organ dysfunction, eventually leading to CVD and CKD [5]. Angiotensin II is involved in the regulation of many organs, mostly through oxidation-reduction reactions generating reactive oxygen species [46] that enhance vascular contraction, vascular cell growth, infiltration of monocytes into the vessel wall [47], vascular dysfunction, inflammation and fibrosis that eventually leads to arterial hypertension and other MS related diseases [46,48]. In the liver, macrophage activation (through systemic signaling) leads to NAFLD progression [48]. In turn, oxidative stress drives to a reduction in antioxidant protective factors produced by the kidneys, such as the Klotho protein (its reduction is considered an early biomarker of CKD progression) [39]. Furthermore, in T2D, uncontrolled hyperglycemia affects renal tubulopathy in early DKD [49]; increased glucose reabsorption in the proximal tubule could cause tubulointerstitial hypoxia and increased oxidative stress [35].

Moreover, multiple events act in parallel including environmental or genetic susceptibilities [50]. Emerging risk factors besides traditional cardiovascular risk factors, such as perturbation of the intestinal microbiota (dysbiosis), are shared by NAFLD, CVD, CKD and other metabolic disorders [39]. A hypercaloric and fiber-poor diet is associated with changes in the intestinal microbiota (dysbiosis) [39]. Intestinal bacteria enhance intestinal permeability, favoring absorption of endotoxins, and produce endogenous ethanol, secondary bile acids [51] and uraemic toxins (e.g., trimethylamine, cresol and indole), which may increase the risk of the development and progression of both NAFLD and CKD, and therefore contributing to CVD development [39].

Regarding genetic susceptibilities in NAFLD, the best-characterized genetic modifier is patatin-like phospholipase domain-containing protein 3 (PNPLA3) [52] encoding I148M (regulator of the mobilization of triglycerides from lipid droplets). PNPLA3 rs738409 polymorphism is associated with NAFLD severity; the PNPLA3 gene is highly expressed in the kidneys and liver pericytes. Renal pericytes are stromal cells that influence angiogenesis and regulate renal medullary and cortical blood flow, promoting renal fibrogenesis and glomerulosclerosis [53]. Sun et al. assessed the effect of PNPLA3 genotypes on biomarkers of renal tubular injury and glomerular function in 217 NAFLD patients who had either normal or abnormal ALT levels [54]. Patients with NAFLD and persistently normal ALT, who carry the PNPLA3 rs738409 G allele, were at higher risk of early glomerular and tubular damage even after adjustment for kidney risk factors and severity of NAFLD histology. Similarly, the presence of the PNPLA3 rs738409 G allele was strongly associated with decreased GFR and intensification of a 24-hour urinary protein excretion in a sample of 142 overweight Italian children/adolescents with biopsy-proven NAFLD, independent of histologic severity, sex, age, measures of adiposity, blood pressure and HOMA-IR [55]. Recently, Mantovani et al. studied 101 post-menopausal T2D women, showing that the G/G genotype of rs738409 in the PNPLA3 gene was strongly associated with lower GFR CKD-EPI and a higher prevalence of CKD even after adjustment for age, duration of diabetes, haemoglobin A1c, HOMA-IR, systolic blood pressure, hypertension treatment and hepatic steatosis [56]. PNPLA3 genotyping may be used to identify individuals with greater susceptibility to NAFLD who are at higher risk of CKD.

In summary (Figure 1), NAFLD exacerbates IR, leading to the release of multiple proinflammatory mediators that promote the activation of the RAS and atherogenic dyslipidemia, key drivers of renal and vascular damage, important in the pathogenesis of both CKD and CVD.

## 5. Clinical Implications: Assessment and Treatment

Clinicians should be aware that patients with T2D and/or MS need liver evaluation, and patients with NAFLD should have a proactive evaluation of all components of the MS [57], particularly assessing the presence of CVD even in the absence of traditional risk factors [58]. Furthermore, clinicians should be aware of the imperative screening of albuminuria in patients with T2D or MS [59]. Without any doubt, patients with non-alcoholic steatohepatitis (NASH) may need early identification and aggressive cardiovascular risk modification. Thus, assessment of liver fibrosis (through LE or fibrosis non-invasive markers) might allow physicians to optimize the timing of appropriate cardiovascular and renal interventions. Scientific studies suggest that LE may be a safe and cost-effective method to evaluate NAFLD and should be considered in the stratification of CV risk. However, in the absence of LE, a simple fibrosis score is independently associated with CVD and CKD, suggesting that fibrosis markers should be considered in primary-care risk assessment. Figure 2 summarizes the assessment and treatment of NAFLD in T2D.

### 5.1. Lifestyle Intervention

The approach to improve liver and cardiovascular outcomes in NAFLD is promptly controlling MS features, including weight loss, insulin sensitization, lipid control and cardiovascular protection. Guidelines [40,58,60] emphasize the importance of modifying lifestyle, including smoke cessation [40,58] and reduced alcohol consumption [40]. The most essential long-term intervention is to recommend a healthy lifestyle that encourages weight loss and control of MS features [41,61]. A weight reduction between 7 and 10% might be sufficient to induce fibrosis regression [62,63] and, therefore, prevent cardiovascular and renal consequences. Bariatric surgery should be considered in patients with a BMI higher than 40 kg/m^2^ or a BMI between 35 and 40 kg/m^2^ with comorbidities [64]. Bariatric surgery may guarantee a 25% weight loss even ten years after surgery [65] with a good long-term control of MS features [1,40,66]. Nonetheless, to guarantee a safe procedure, bariatric surgery should be performed in high-volume hospitals with the involvement of a multidisciplinary team [67].

Regardless of weight loss, improved macronutrient composition [45] and physical exercise [1,60] may act independently to prevent NAFLD progression. Performing between 150–250 min/week of moderate aerobic exercise is suggested [68], although exercising for more than 250 min/week may warrant a significant metabolic improvement [1]. High-intensity intermittent exercises have also proven to be beneficial in NAFLD [69]. Regarding dietary intervention, the Mediterranean diet has shown significant beneficial effects on blood pressure [70], glucose and lipid metabolism [71,72], and therefore improves NAFLD [45,58,73]. The excessive consumption of refined and simple carbohydrates, saturated and trans-fat, animal protein and low fiber intake is associated with NAFLD progression [73].

Lifestyle intervention is the key therapeutic approach. However, pharmacological therapies that improve blood pressure, glucose and lipid metabolism are essential.

### 5.2. Specific Management of Arterial Hypertension in NAFLD

Close monitoring with adjusted antihypertensive drug therapy to reach blood pressure targets is needed, particularly in hypertensive patients with NASH and/or progressive hepatic fibrosis [74]. Inhibition of RAS has shown to be effective in reducing cardiovascular disease and the progression of DKD [75]. As established in the pathophysiology, RAS is essential in NAFLD pathophysiology. Therefore, angiotensin-converting enzyme (ACE) inhibitors and angiotensin II receptors blockers (ARB) influence cytokine production [76]. The RAS hepatic effect is mediated by angiotensin II receptor type I, which mediates the actions of angiotensin-II in hepatocytes, bile duct cells, hepatic stellate cells and vascular endothelial cells [77]. Consequently, the inhibition of RAS may improve intracellular signaling pathways.

In a systematic review including eleven trials (*n* = 66,608 patients) and in a meta-analysis including 12 randomized controlled clinical trials (*n* = 116,220 patients, of whom 72,333 did not have diabetes at baseline), ACE inhibitors were demonstrated to reduce the incidence of T2D [78,79]. Interestingly, in an animal model, ACE inhibitors revealed a reduction of liver fibrosis [80,81]. However, the influence of the ACE inhibitors in NAFLD is limited. On the other hand, through reduction of IR [82], ARB demonstrated to reduce the incidence of T2D [79] and also sustained efficacy in patients with NAFLD [83,84]. After six months, olmesartan and telmisartan significantly improved IR and transaminase levels [82]. In a pilot study including 54 patients with biopsy-proven NASH, telmisartan and valsartan revealed an improvement in fibrosis stage [85]. Losartan has been evaluated in small studies showing improvement in hepatic necroinflammation and fibrosis [86,87].

### 5.3. Specific Lipid Control in NAFLD

Lipid control is a crucial intervention in NAFLD and T2D for primary and secondary prevention of CVD [88,89]. Although NAFLD progression reduction was suggested through the use of statins [90,91,92,93], guidelines strongly recommend its prescription to prevent CVD [58]. Statins can be safely used in NAFLD [94,95]; however, discontinuation should be considered if ALT levels rise more than three times the upper limit normal range value [58,96]. Similarly, ezetimibe [97,98], fibrates [99,100], Omega-3 [101] and PCSK9 inhibitors [102] have a recognized effect on lipid modulation.

### 5.4. Specific T2D Treatment in NAFLD

Glycemic control is crucial to prevent T2D complications. Although pharmacological treatment has not yet been approved by international agencies for the treatment of NAFLD, antidiabetic drugs with known positive effects on liver fibrosis are recommended in T2D and NAFLD [40,58,103] to prevent NAFLD progression [58,61,68] and, therefore, CVD and CKD. The antidiabetic drugs with proven efficacy in NAFLD are pioglitazone, glucagon-like peptide-1 (GLP-1) analogs and sodium-glucose cotransporter 2 (SGLT-2) inhibitors [20,40,58,61,104]. Figure 3 summarizes the histological liver effect and cardiovascular recommendation of antidiabetic drugs in NAFLD.

Pioglitazone improves insulin sensitivity and mitochondrial dysfunction in hepatocytes [41]. A randomized controlled trial including 55 patients with prediabetes/T2D and NASH showed a significant decrease in inflammation, ballooning necrosis and steatosis in the group treated with pioglitazone (45 mg/day during 6 months) [105]. Nevertheless, in the daily clinical practice, clinical practitioners are hesitant to prescribe pioglitazone due to its known side effects (e.g., heart failure decompensation, controversial relationship with bladder cancer, bone mineral density reduction, weight loss) [106,107]. Nevertheless, a meta-analysis including 516 patients confirmed the efficacy and safety of pioglitazone in the management of NAFLD [108]. Additionally, pioglitazone was proven to reduce the incidence of stroke and myocardial infarction in patients with a previous ischemic stroke [109].

GLP-1 agonists reduce hepatic glucose production, promote insulin secretion and decrease postprandial glucagon levels in a glucose-dependent manner, and induce satiety and weight loss [110]. GLP-1 agonists are approved for the treatment of T2D and can be safely used in CKD. Moreover, GLP-1 agonists prevent the onset of macroalbuminuria and decelerate the decline of GFR in diabetic patients [111]. The LEAN trial evidenced a histological benefit in 52 patients treated with liraglutide vs. placebo for 48 weeks (resolution of NASH: 39% vs. 9%) [112]. The LEAD program performed an individual patient data metanalysis where liver enzyme reduction was significantly achieved [113]. Similar effects were reported with other GLP-1 agonists [107]. A trial of exenatide vs. insulin therapy during eight weeks was associated with greater reversal of liver fat (assessed through ultrasound) [114], and similar results were recently reported after 24 weeks of treatment, although liver fat was assessed through MR spectroscopy [115]. GLP-1/glucose-dependent insulinotropic peptide receptors dual agonist is a novel agent that showed NASH improvement in mice [116,117] and significantly decreased fibrosis biomarkers and increased adiponectin in patients with T2D [118]. Dulaglutide [119], liraglutide [120] and semaglutide [121] were demonstrated to have protective cardiovascular effects outside of glycemic control.

In the kidney, SGLT-2 is the main cotransporter involved in glucose reabsorption [122]. Data from the use of SGLT-2 inhibitors in NAFLD are scarce. SGLT-2 inhibitors have demonstrated liver enzymes improvement, with additional beneficial effects on various metabolic parameters in T2D patients with NAFLD [123,124]. A meta-analysis including 20 studies (*n* = 1950 patients with T2D, with or without NAFLD, who were treated with SGLT-2 inhibitors for at least 8 weeks) evidenced a significant decrease in liver steatosis compared with placebo or other oral antidiabetic drugs in patients with T2D [125]. Interestingly, a recent double-blind trial assigned 106 patients with NAFLD and T2D to receive empagliflozin 10 mg (*n* = 35), pioglitazone 30 mg (*n* = 34) or placebo (*n* = 37) for 24 weeks. Empagliflozin, in contrast to pioglitazone, was associated with improvement in liver steatosis and fibrosis (evaluated through LE) in patients with NAFLD and T2D [126]. In Japan, ipragliflozin was compared to pioglitazone, showing a similar effect on hepatic fat content (measured through CT) [127]. Likewise, luseogliflozin was shown to be superior to metformin in reducing hepatic fat content (measured through CT) [128]. There is still no evidence evaluating SGLT-2 inhibitors’ effect on liver histology, but they are medications with encouraging results due to their potential to promote weight reduction, improve glycemic control [129], reduce CVD (canagliflozin, dapagliflozin and empagliflozin) and reduce the progression of CKD in patients with and without diabetes [130,131].

## 6. Conclusions

NAFLD pathogenesis is closely related to that of both CVD and CKD, sharing different mechanisms including IR, oxidative stress, chronic and systemic micro-inflammation and genetic predisposition. Patients with NAFLD, T2D, CVD and/or CKD should also be assessed for NASH and advanced fibrosis in order to identify those at the highest risk of adverse outcomes so that appropriate management strategies can be implemented. There is a need for large prospective collaborative studies to understand the clinical and prognostic implications better, including mortality, CVD and CKD progression. Moreover, there is a need to include renal outcomes in future randomized controlled trials focused on testing the efficacy and safety of novel treatments for NAFLD in T2D.

## Figures and Tables

**Figure 1 jcm-10-02040-f001:**
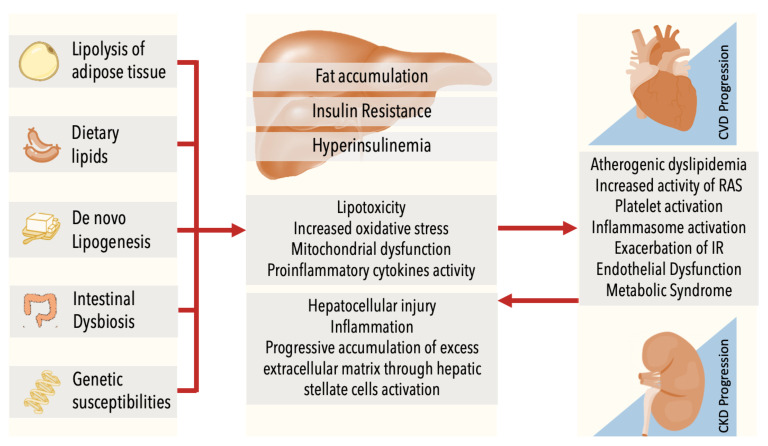
NAFLD, CKD and CVD share common risk factors. Lipolysis of adipose tissue, dietary lipids, de novo lipogenesis, dysbiosis and genetic susceptibilities act in parallel, contributing to fat accumulation. It leads to insulin resistance (IR), enhancing liver trygliceride accumulation and results in compensatory hyperinsulinemia which increases the hepatic fatty acid uptake, alters triglycerides transportation and inhibits liver β-oxidation. There is an intensification of the pro-inflammatory cytokine activity that is associated with oxidative stress-mediated lipotoxicity, increased activity of RAS, platelet activation, inflammasome activation and mitochondrial dysfunction. These processes contribute to further exacerbation of IR, hepatocellular injury, inflammation and the progressive accumulation of excess extracellular matrix through hepatic stellate cells activation. NAFLD: Non-alcoholic Fatty Liver Disease, CKD: Chronic Kidney Disease; CVD: Cardiovascular Disease; IR: insulin resistance; RAS: renin-angiotensin-aldosterone system.

**Figure 2 jcm-10-02040-f002:**
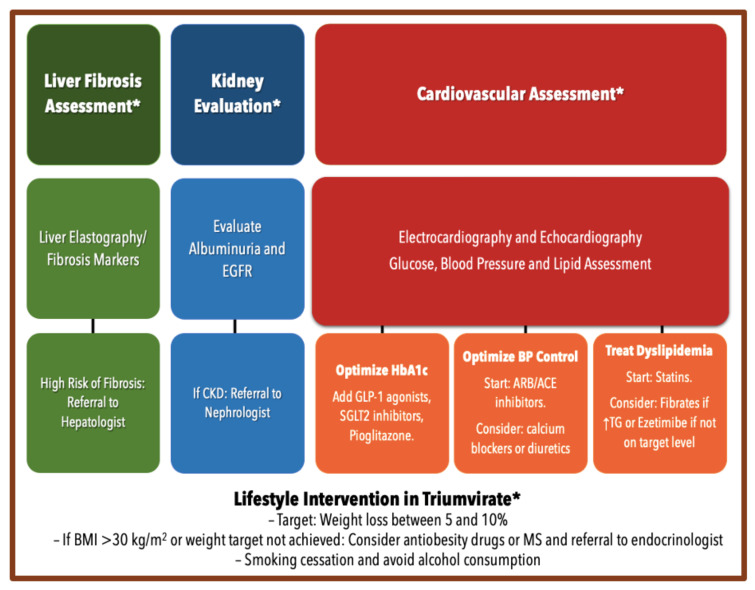
Assessment and Treatment of Non-alcoholic Fatty Liver Disease in Type 2 Diabetes. ACE: Angiotensin converting enzyme; ARB: Angiotensin-II receptor antagonists or blockers; BMI: body mass index; EGFR: estimated glomerular filtration rate; GLP-1: Glucagon-like Peptide-1; MS: Metabolic surgery; SGLT2: Sodium-glucose cotransporter 2; TG: Triglycerides; * Triumvirate Lifestyle Intervention.

**Figure 3 jcm-10-02040-f003:**
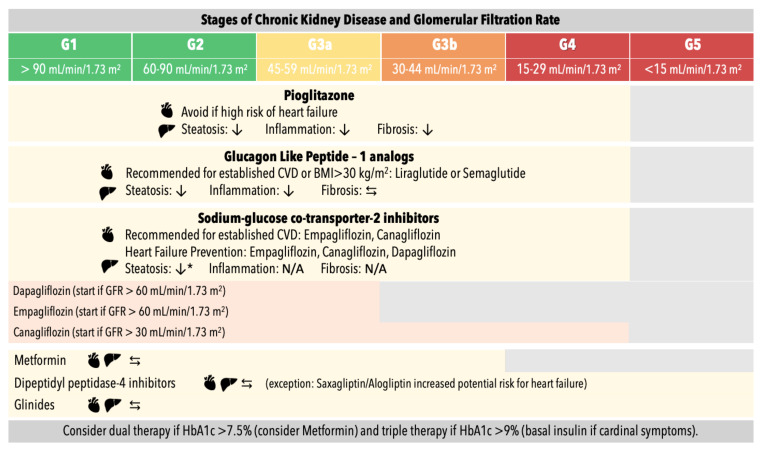
Antidiabetic drugs in patients with type 2 diabetes, chronic kidney disease and non-alcoholic fatty liver disease: description of histological liver effects and cardiovascular recommendation. ↓: reduce; ⇆: neutral effect; N/A: no data available; * No histological evidence, GFR: glomerular filtration rate.

## Data Availability

Not applicable.

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
