# Peer review of "Diabetic Kidney Disease, Cardiovascular Disease and Non-Alcoholic Fatty Liver Disease: A New Triumvirate?"

_jcm, 2021, doi:10.3390/jcm10092040_

Round 1
Reviewer 1 Report
The authors present a review discussing the current state of knowledge for the overlap of risk and development for nonalcoholic fatty liver disease, chronic kidney disease, and cardiovascular disease. In general, the authors are to be commended. The review is well written and well cited. However, this reviewer feels that while the points of discussion are very relevant to the topic, the literature review with regard to CVD is a little “light.”
- Lines 158-160: “RAS activation has been implied in the production of pro-inflammatory cytokines aggravating oxidative stress and supporting subclinical organ dysfunction, eventually leading to CKD…”
There is a very solid body of evidence showing that this is also true of CVD. These mechanisms are also later echoed in lines 276-284, so I think is worth at least some discussion in the context of the mechanisms shared by this triumvirate. In addition to its role in RAAS, angiotensin II can directly promote the expression of NADPH oxidases (PMID: 22530599) to cause oxidative stress and vascular injury. Angiotensin II is also a key signaling molecule for leukocyte extravasation into the endothelium of vessels, wherein circulating monocytes then differentiate and cause a pro-oxidative, pro-inflammatory environment which can be the initiating factor for CVD (PMID: 21875910, amongst others) - Is it possible that this recruitment from circulation is also relevant in the context of NAFLD? (PMID: 30405618, 33236445) I think this would elegantly link some of the data discussed.
- Line 332: Liraglutide has also been demonstrated to have protective cardiovascular effects outside of glycemic control (PMID: 27295427, LEADER trial. PMID: 31747801 for mechanistic insights related to those mentioned above).
Very minor:
- Lines 74, 122: Green text?
Author Response
Response to Reviewer 1 Comments
Thank you for your comments. The manuscript has been edited following each of your suggestions:
Point 1: Lines 158-160: “RAS activation has been implied in the production of pro-inflammatory cytokines aggravating oxidative stress and supporting subclinical organ dysfunction, eventually leading to CKD…”
There is a very solid body of evidence showing that this is also true of CVD. These mechanisms are also later echoed in lines 276-284, so I think is worth at least some discussion in the context of the mechanisms shared by this triumvirate. In addition to its role in RAAS, angiotensin II can directly promote the expression of NADPH oxidases (PMID: 22530599) to cause oxidative stress and vascular injury. Angiotensin II is also a key signaling molecule for leukocyte extravasation into the endothelium of vessels, wherein circulating monocytes then differentiate and cause a pro-oxidative, pro-inflammatory environment which can be the initiating factor for CVD (PMID: 21875910, amongst others)
Comment 1: We have modified Lines 158-160 to: “RAS activation has been implied in the production of pro-inflammatory cytokines aggravating oxidative stress and supporting subclinical organ dysfunction, eventually leading to CVD and CKD”. In the lines ahead, we made a broader description of RAS effect on both entities.
Point 2: Is it possible that this recruitment from circulation is also relevant in the context of NAFLD? (PMID: 30405618, 33236445) I think this would elegantly link some of the data discussed.
Comment 2: RAS activation implication in CVD and CKD was added to the pathophysiology trough lines 160-165. PMID: 22530599, PMID: 21875910 and PMID: 30405618 were added to the references.
Point 3: Line 332: Liraglutide has also been demonstrated to have protective cardiovascular effects outside of glycemic control (PMID: 27295427, LEADER trial. PMID: 31747801 for mechanistic insights related to those mentioned above).
Comment 3: Line 332 was modified to: “Dulaglutide[116], liraglutide[117] and semaglutide[118] have demonstrated to have protective cardiovascular effects outside of glycemic control.”
Point 4: Lines 74, 122: Green text?
Comment 4: We have accepted the changes in line 74 and in line 122.
Reviewer 2 Report
NAFLD is a growing concern all over the world due to its increasing prevalence and associated complications. The authors did a great job in writing up this comprehensive review article explaining the pathophysiology of NAFLD, its association with CKD, CVD and T2DM, and the available interventions. The images are appropriately used and easy to comprehend.
Author Response
General Comments. NAFLD is a growing concern all over the world due to its increasing prevalence and associated complications. The authors did a great job in writing up this comprehensive review article explaining the pathophysiology of NAFLD, its association with CKD, CVD and T2DM, and the available interventions. The images are appropriately used and easy to comprehend.
General Response: Thank you for your comments.
Reviewer 3 Report
the authors focused on connection between NAFLD, CKD and cardiovascular disease. The subject is not full of novelty but it is well written and described.
There are some minor issue need to be addressed
1- Epicardial Fat is subsequently indicated as EF. I suggest to change the acronym due it may be easily confused with the more used Ejection Fraction
2- GFR is not indicate in meaning. I suggest to write the complete meaning of the acronym the first time it is named in the text
3- No indication about the meaning about the meaning of G1-5 in Figure 3 neither in text is indicate. I guess it is the grade of CKD but a specific indication may be useful
Finally, there are some typing errors (i.e. at line 333) that need to be addressed
Author Response
Thank you for your comments. The manuscript has been edited following each of your suggestions:
Point 1: Epicardial Fat is subsequently indicated as EF. I suggest to change the acronym due it may be easily confused with the more used Ejection Fraction
Comment 1: We have modified the acronym for epicardial fat to EAT (epicardial adipose tissue).
Point 2: GFR is not indicate in meaning. I suggest to write the complete meaning of the acronym the first time it is named in the text
Comment 2: We have added the acronym GFR and its meaning on line 120.
Point 3: No indication about the meaning about the meaning of G1-5 in Figure 3 neither in text is indicate. I guess it is the grade of CKD but a specific indication may be useful
Comment 3: We have indicated the meaning of G1-G5 in the figure and in the text of Figure 3.
Point 4: Finally, there are some typing errors (i.e. at line 333) that need to be addressed
Comment 4: Line 333 (now, Line 340) was revised and modified from: “SGLT-2 is the major cotransporter involved in glucose reabsorption in the kidney [119]. SGLT-2 inhibitors have evidenced liver enzymes improvement, with additional beneficial effects on various metabolic parameters in T2D patients with NAFLD[120][121]. “, to: “In the kidney, SGLT-2 is the main cotransporter involved in glucose reabsorption[122]. SGLT-2 inhibitors have demonstrated liver enzymes improvement, with additional beneficial effects on various metabolic parameters in T2D patients with NAFLD[123][124].”
Reviewer 4 Report
- According to the title and the aim of the study, the authors shound focus only to the coexistence of these diseases. Therefore, they should focus both from the pathophysiology point of view and for the treatment only in the connective data for the three diseases. It is means that they put data for diabetic patients with CKD and not generally for T2DM.
- For example, for the management of diabetes with CKD, agents that are largely cleared by the kidnays, shoujd be avoided, whereas agents metabolized by the liver and/or partially excreted by the kidneys, may require dose reduction or discontinuation, particulrly when eGFR falls bellow 30.ML/MIN/1.73m2.
- Only pioglitagone and vit E are currently prescribed .Liragglutide and semaglutide have gained attention.SGLT-2 inhibitors, have suggested cacardiovascular benefit, whih may extent to patients with CKD, but we have no data conserning NAFLD.
- Are there data for the possible relatioship in diabetics with various stages of CKD?
Author Response
Thank you for your general comments. Our responses would be the followings:
Point 1: According to the title and the aim of the study, the authors shound focus only to the coexistence of these diseases. Therefore, they should focus both from the pathophysiology point of view and for the treatment only in the connective data for the three diseases. It is means that they put data for diabetic patients with CKD and not generally for T2DM.
Comment 1: We are grateful for the recommendations made by the reviewer. Since this is a multifactorial pathology, we have found it necessary to take a broader and more general approach that allows understanding in its proper framework. The three pathologies are closely related to each other, but they are also considered among the main causes of morbidity and mortality in the world, while at the same time they are associated with a wide range of other disorders, so we considered it necessary for the reader to have a complete view of each entity in order to comprehend their correlation. In addition to this, this review do not seek to refer to patients in whom the three pathologies coincide, but rather to the existing relationship in terms of pathophysiological aspects that change the evolution and prognosis of each of the pathologies.
Point 2: For example, for the management of diabetes with CKD, agents that are largely cleared by the kidnays, shoujd be avoided, whereas agents metabolized by the liver and/or partially excreted by the kidneys, may require dose reduction or discontinuation, particulrly when eGFR falls bellow 30.ML/MIN/1.73m2.
Comment 2: We fully agree with the reviewer. In the management of patients with CKD and/or altered liver function, it is important to take into account the dose adjustment to avoid pharmacological toxicity. However, each case depends on the pharmacological group used and the clinical situation of each patient. We appreciate your comment because the key is indeed in a personalized approach to each clinical situation.
Point 3: Only pioglitagone and vit E are currently prescribed .Liragglutide and semaglutide have gained attention.SGLT-2 inhibitors, have suggested cacardiovascular benefit, whih may extent to patients with CKD, but we have no data conserning NAFLD.
Comment 3: We deeply agree with the reviewer that conclusive clinical trials on drugs in NASH are still lacking, nevertheless recent observational studies (i.e. Shao et al. 2020) show benefits of iSLT2 and aGLP-1 in NAFLD. However, recent studies are needed.
Point 4: Are there data for the possible relatioship in diabetics with various stages of CKD?
Comment 4: The review includes all the recent scientific literature on the subject. The studies conducted include various stages of CKD according to the population analyzed.
Round 2
Reviewer 4 Report
I have read the answers from my comments.
- Point 1. The authors have to mention in the paper that they put a general approach and not a precise approach of the three diseases.
- Point 3. The authors have to clarify that data from the use of SGLT-2 inhibitors in NAFLD patients are very little in order to discuss for benefits.
Author Response
Thank you for your comments. The manuscript has been edited following each of your suggestions:
Point 1. The authors have to mention in the paper that they put a general approach and not a precise approach of the three diseases.
Comment 1: We added on Line 40: “This review intends to give a general approach to these three diseases and not a specific approach for each condition.”
Point 2. The authors have to clarify that data from the use of SGLT-2 inhibitors in NAFLD patients are very little in order to discuss for benefits.
Comment 3: We added on Line 391: “Data from the use of SGLT-2 inhibitors in NAFLD is scarce.”